# Fault Diagnosis of Permanent Magnet Synchronous Motor Based on Stacked Denoising Autoencoder

**DOI:** 10.3390/e23030339

**Published:** 2021-03-12

**Authors:** Xiaowei Xu, Jingyi Feng, Liu Zhan, Zhixiong Li, Feng Qian, Yunbing Yan

**Affiliations:** 1School of Automobile and Traffic Engineering, Wuhan University of Science and Technology, Wuhan 430081, China; fengjingyi97@gmail.com (J.F.); zhanliu2021@163.com (L.Z.); feng.qian@wust.edu.cn (F.Q.); yanyb@126.com (Y.Y.); 2Yonsei Frontier Lab, Yonsei University, 50 Yonsei-ro, Seodaemun-gu, Seoul 03722, Korea; zhixiong.li@yonsei.ac.kr

**Keywords:** stacked denoising autoencoder, permanent magnet synchronous motor, support vector machine, fault diagnosis

## Abstract

As a complex field-circuit coupling system comprised of electric, magnetic and thermal machines, the permanent magnet synchronous motor of the electric vehicle has various operating conditions and complicated condition environment. There are various forms of failure, and the signs of failure are crossed or overlapped. Randomness, secondary, concurrency and communication characteristics make it difficult to diagnose faults. Meanwhile, the common intelligent diagnosis methods have low accuracy, poor generalization ability and difficulty in processing high-dimensional data. This paper proposes a method of fault feature extraction for motor based on the principle of stacked denoising autoencoder (SDAE) combined with the support vector machine (SVM) classifier. First, the motor signals collected from the experiment were processed, and the input data were randomly damaged by adding noise. Furthermore, according to the experimental results, the network structure of stacked denoising autoencoder was constructed, the optimal learning rate, noise reduction coefficient and the other network parameters were set. Finally, the trained network was used to verify the test samples. Compared with the traditional fault extraction method and single autoencoder method, this method has the advantages of better accuracy, strong generalization ability and easy-to-deal-with high-dimensional data features.

## 1. Introduction

As one of the important parts of an electric vehicle, the permanent magnet synchronous motor (PMSM) has the advantages of small volume, high-efficiency and high-power density. However, most electric vehicle motors work in closed, narrow, complex and harsh environments [1,2]. Under the combined action of electric field force and magnetic field force, the load changes greatly, and a variety of faults are prone to occur, such as stator inter-turn winding short circuit [3], shafting misalignment, permanent magnet loss of excitation and so on [4]. The occurrence of one kind of fault may induce the occurrence of another kind of fault and even lead to the coupling effect of various faults. The generation of coupling fault will cause irreversible damage to the performance of the motor itself, especially in the high-temperature working environment, which will seriously affect the normal operation of the motor and electric vehicle. Therefore, the fault diagnosis analysis of electric vehicle PMSM is of great significance to the development of electric vehicles and motors [5].

At present, many scholars have carried out fault diagnosis and analysis of permanent magnet synchronous motor. In reference [6], the positive envelope of bus current and three-phase current is taken as the signal extraction object, and the wavelet packet algorithm is used as the bearing fault signal extraction method to identify the bearing fault information in the DC motor. In reference to [7], the multi-loop mathematical model of Asynchronous motor is established according to the circuit method. The finite element analysis method is used to simulate the motor with multiple fault types, and the obtained current spectrum is analyzed to verify the fault mechanism characteristics of the motor. In reference [8], the mathematical model of the motor is established in MATLAB/Simulink to simulate various faults. The current, torque and speed waveforms of the motor under normal operation are compared and analyzed, and the basic characteristics information of the permanent magnet synchronous motor faults are obtained. However, the traditional identification methods of motor fault diagnosis based on a mathematical model and electrical signal are highly dependent on the accuracy of the model, and the selection of signal wave base has certain limitations, so the accuracy of motor fault feature extraction and analysis still needs to be improved.

At present, with the continuous development of artificial intelligence, machine learning has been cross-applied in various fields [9], such as the recovery and prediction of missing data [10], the judgment of stock price changes [11], and the detection of urban road obstacles [12]. These areas span biology, medicine, machinery, finance, etc.; [13,14,15,16], which has become the future development trend. The deep learning algorithm proposed by Hinton [17] and others is increasingly used in the fields of pattern recognition and deep feature extraction [18], which has a good application prospect in the faults detection of various mechanical equipment. In view of the inaccuracy of traditional manual fault feature extraction of vehicle motor [19], large capacity of fault data, many types of data and slow transmission speed [20], multilevel network analysis structure and adaptive learning process can be used to extract faults data features more accurately [21,22,23]. Reference [24] constructs a transformer fault recognition framework based on block training of the Adaboost-RBF algorithm. Reference [25] uses the wavelet packet analysis method to extract features of vibration data collected from motor experiments and then inputs the decomposed data as test samples into the support vector machine for classification diagnosis. Reference [26] adopts the sparse autoencoder algorithm to extract features of motor bearing vibration signal so as to achieve fault diagnosis. Reference [27], the denoising autoencoder is used to extract the features of the aero-engine gas path fault and combined with FBRF classifier. The features extracted for aero-engine fault analysis have good robustness. However, in the actual motor experiments, the collected data often have noise error compared with the real value and cannot accurately collect all the monitored signal values. Most of the above methods using deep neural networks for mechanical equipment faults detection do not consider the data with noise, and the processing of interference signal is weak.

As an unsupervised learning algorithm, autoencoders (AE) can accurately learn the internal characteristics of complex signals from unlabeled data, which has obvious advantages for high-dimensional motor data processing [28]. Based on the above research on autoencoder and the application of deep learning method in various fields, this paper proposes a fault diagnosis method based on the stacked denoising autoencoder (SDAE) algorithm for permanent magnet synchronous motor (PMSM) used in an electric vehicle. This method is mainly composed of two parts: first, it uses SDAE to extract the features of the collected operation data of PMSM and then inputs the extracted data into the support vector machine (SVM) for classification calculation, so as to identify the motor fault types, and finally achieve the purpose of fault detection. The main purpose of the SDAE algorithm is to extract the features of the collected PMSM data, which is equivalent to label the data and facilitate the subsequent SVM to classify it. Moreover, the SDAE used in this paper adds noise processing in the data input, which has better adaptability for the actual incomplete data.

The paper will discuss the proposed method according to the following aspects: in the second part, the principle of a single AE algorithm is introduced first, and gradually the working principle of the SDAE and SVM algorithm is introduced; in the third part, the specific steps of motor diagnosis method proposed in this paper are described; in the fourth part, the feasibility of the diagnosis method proposed in this paper is verified by the bearing data set, and then the motor operation data obtained from the experiment is used for practical verification. The experimental results show that the method has better accuracy and faster running speed. Finally summarizes the whole paper. This method can enhance the self-adaptive diagnosis ability of stacked denoising autoencoder by artificially adding noise to the input data to simulate the damage data collected and can effectively extract and classify the motor fault features.

## 2. Principle

The fault diagnosis of permanent magnet synchronous motor (PMSM) based on the stacked denoising autoencoder algorithm includes two parts: stacked denoising autoencoder and support vector machine (SVM) classification. A stacked denoising autoencoder can be regarded as the superposition of multiple denoising autoencoders. The collected motor running signals are taken as the input. The input data are randomly set to 0 or 0.2–0.3 times of Gaussian noise is added to simulate the damaged data to predict the output result of the original undamaged data. The fault characteristics are obtained by minimizing the reconstruction error. Finally, the nonlinear transformation of the support vector machine is used to classify the extracted fault features and output the final results.

### 2.1. Autoencoder Network Principle

An autoencoder is a kind of neural network proposed by Yann Lecun, which can realize the BP backpropagation algorithm. After training, it can copy the input to the output. The autoencoder can analyze the data characteristics well by reducing the dimension of the data, so it is often used to realize the functions of data anomaly analysis, data denoising, image analysis and data retrieval. Figure 1 shows the network structure of the single hidden layer autoencoder, in which the coding layer encodes the input data as the representation, while the decoding layer decodes the representation as to the output with a minimum loss, and the reconstruction error is the basis for measuring the learning effect [29].

There are n groups of training samples *X* = {*X*(1), *X*(2), *X*(3), …, *X*(N)}, each group of samples is an *n*-dimensional vector, the coding process can be expressed as:(1)h=f(x)=S(Wx+b)

Among them, hidden layer *h*{*h*1(1),*h*2(1),*h*3(1),…, *hm*(1)} is the m-dimensional vector (*m* ≥ 1), which can be regarded as m neurons, *W* is the weight matrix of order *m × n*, *b* is the hidden layer bias vector of dimension *m*, *S* is the sigmoid activation function.

The decoding process can be expressed as follows:(2)$$\hat{x}=g\left(h\right)=S\left(W^{\prime }h+b^{\prime }\right)$$
where *W′* is the weight matrix of order *m × n* and *b′* is the output bias vector of dimension *m*. For the purpose of minimizing the reconstruction error of autoencoder, let the reconstruction error be: *J*(*θ*), *θ* = {*W*,*b*}, $J\left(\theta \right)=\frac{1}{N}\sum_{1}^{N}loss\left(x,\hat{x}\right)$, where the loss function is the loss function of the reconstruction error. In this paper, the mean square error method is used to calculate:(3)$$Loss\left(x,\hat{x}\right)=\frac{1}{2}|\left|\hat{x}-x\right|{|}^{2}+\frac{\lambda }{2}\left(\left|\left|W\right|{|}^{2}+\right|\left|W^{\prime }\right|{|}^{2}+\left|\left|b\right|{|}^{2}+\right|\left|b^{\prime }\right|{|}^{2}\right)$$
where *λ* is the weight adjustment coefficient. After iterative training, the reconstruction error can reach a smaller value, and the accuracy of data feature extraction can be improved.

### 2.2. Stacked Denoising Autoencoder Structure

In order to avoid the overfitting phenomenon of autoencoder in the process of data processing, the denoising autoencoder (DAE) adds “damage noise” or random zeroing to the input data on the basis of the original simple autoencoder to simulate the input of damaged data, which can effectively improve the robustness of model learning [30].

Figure 2 is the network structure diagram of the denoising autoencoder. In the input part, the original data becomes the missing or damaged impurity data after a specific destruction processing as the new input, and the input replaces the original data for automatic coding and learning process. In this case, the minimization objective function of denoising autoencoder becomes:(4)$$Loss\left(x,g\left(f\left(\tilde{x}\right)\right)\right)$$

Among them is the processed damage input. Generally, a Gaussian noise or dropout method can be used to denoise the original data.

Due to the limitations of the single-layer network model for complex data processing, multiple denoising autoencoders are introduced to form a stacked Denoising autoencoder. The input and output of each layer can be seen as a separate network structure. Equations (5) and (6) are the encoding and decoding process of the SDAE layer 1:(5)$$h=f\left(\tilde{x}\right)=S\left(W_{1}\tilde{x}+b_{1}\right)$$
(6)$$\hat{x}=g\left(h\right)=S\left(W_{1}^{\prime }h+b_{1}^{\prime }\right)$$

After the first layer training, the weight *W* and bias *b* are updated by gradient descent method:(7)$$W_{2}=W_{1}-\eta \frac{\partial Loss}{\partial W_{1}}$$
(8)$$W_{2}^{\prime }=W_{1}^{\prime }-\eta \frac{\partial Loss}{\partial W_{1}^{\prime }}$$
(9)$$b_{2}=b_{1}-\eta \frac{\partial Loss}{\partial b_{1}}$$
(10)$$b_{2}^{\prime }=b_{1}^{\prime }-\eta \frac{\partial Loss}{\partial b_{1}^{\prime }}$$

The output of the first hidden layer can be used as the input of the next layer to continue the iterative training until the training of all layers is completed. When there are L hidden layers in common, the denoising autoencoder network of each layer can be expressed as:(11)$$h_{l}=f\left({\tilde{x}}_{l-1}\right)=S_{l}\left(W_{l}{\tilde{x}}_{l-1}+b_{l}\right)$$
(12)$${\hat{X}}_{l-1}=g\left(h_{l}\right)=S_{l}\left(W_{l}^{\prime }h_{l}+b_{l}^{\prime }\right)$$

Through the single-layer successive updating transformation of the features by stacked denoising autoencoder, the high-dimensional feature expression of the data can be realized, and the method has better robustness and accuracy for the feature extraction of the data.

### 2.3. SVM Classifier

As a supervised learning algorithm, the support vector machine mainly uses the idea of the maximum interval to solve the problem of data classification in the field of pattern recognition. It can be regarded as an optimization algorithm for solving convex quadratic programming and has a good classification effect for both linear and nonlinear problems. Compared with the deep learning classification method, SVM is easy to operate in the program, and it can get higher accuracy without using much data, which is suitable for the situation of fewer motor data collected in this paper, and it is not easy to appear overfitting phenomenon. Furthermore, the addition of kernel function makes SVM can accurately reflect the nonlinear characteristics, so this paper selects SVM as the classifier of motor diagnosis research.

In the SVM algorithm, let the training data set in the given feature space be: *T* = {$\left(x_{1},y_{1}\right),\left(x_{2},y_{2}\right),\ldots ,\left(x_{n},y_{n}\right)$}, $x_{i}\in R_{n}$, $y_{i}\in \left\{+1,-1\right\}$, *i* = 1,2,…,*n*, $x_{i}$ is the *i*-th eigenvector and $y_{i}$ is the $x_{i}$ class marker. The separation hyperplane equation is:(13)$$\omega x_{i}+b=0$$
where ω is the weight vector and *b* is the offset. The constrained optimization problem of separating hyperplane and classification decision function by maximum interval can be transformed into solving the minimum value of the following equation:(14)$$\frac{1}{2}(\omega^{T}\omega )+C\overset{n}{\underset{i=1}{\sum }}\zeta_{i}$$

The constraint conditions of Equation (14) are as follows: $y_{i}\left(\omega^{T}\omega +b\right)\geq 1-\zeta_{i},\;\zeta_{i}\geq 0$. Where *C* is the penalty coefficient and $\zeta_{i}$ is the relaxation coefficient. By introducing Lagrange function and taking Radial Basis Function (RBF) *K*$\left(x_{i},x_{j}\right)$ as the inner product kernel function of the algorithm, the interval maximization problem can be obtained:(15)$$Q\left(\alpha \right)=\overset{n}{\underset{i=1}{\sum }}\alpha_{i}-\frac{1}{2}\overset{n}{\underset{i,j=1}{\sum }}\alpha_{i}\alpha_{j}y_{i}y_{j}\left(x_{i},x_{j}\right)$$
where $\alpha_{i}$ is the Lagrange multiplier corresponding to $x_{i}$. And the constraint condition of Equation (16) is as follows: $\overset{n}{\underset{i=1}{\sum }}\alpha_{i}y_{i}=0,\;\alpha_{i}\geq 0,\;i=1,2,\ldots ,n$. The decision function is as follows:(16)$$f\left(x\right)=sgn\left(\overset{n}{\underset{i=1}{\sum }}\alpha_{i}y_{i}K\left(x_{i},x_{j}\right)+b\right)$$

According to the principle of statistics, the accuracy of the SVM classifier can be expressed as the ratio of the number of samples correctly classified on a given test set to the total number of samples, as shown in Equation (17):(17)$$R_{test}=\frac{1}{N^{\prime }}\overset{N^{\prime }}{\underset{i=1}{\sum }}I\left(y_{i}=\hat{f}\left(x_{i}\right)\right)$$
where *N’* is the capacity of training sample; I is the indicator function, when *y* ≠ $\hat{f}\left(x_{i}\right)$, *I* = 1; Otherwise, *I* = 0.

## 3. Method

As an electromechanical coupling component, a permanent magnet synchronous motor (PMSM) often works in a harsh working environment during its operation. After the motor has a tendency to damage, the coupling effect of the electromagnetic field will make the faults happen faster and more obvious, especially inter-turn short circuits. For example, when the motor has a slight short circuit fault, if it is not checked and repaired in time, it will lead to an increase in the motor operating temperature. That will cause the change of the working temperature field, which will increase the degree of the motor inter-turn short circuit fault, and even cause the demagnetization of the permanent magnet. Moreover, the change of the working magnetic field caused by demagnetization will aggravate the degree of short circuit and other faults eventually.

The original feature extraction method uses artificial discrimination to operate, the accuracy of the diagnosis results depends on the technical level and practical experience of the diagnosis personnel, its self-learning ability is weak, and the intelligence level is low, while the fault diagnosis method of signal analysis has a strong dependence on data and poor generalization ability. Aiming at the shortcomings of traditional manual or signal processing methods, such as the inaccurate and slow speed of motor fault feature extraction, this paper proposes a method of feature extraction of permanent magnet synchronous motor fault using the SDAE method according to the principle of stacked denoising autoencoder, and combined with support vector machine to complete the classification of fault features. Figure 3 shows the main process of motor fault diagnosis based on stacked denoising autoencoder.

### 3.1. SDAE Diagnostic Process

In the stacked denoising autoencoder part, the data processing generally includes the following steps:The vibration and speed signals collected from the PMSM fault experiment are divided into training samples and test samples, and the data samples of known fault types are packaged to establish the motor fault signal database;The vibration data are normalized and preprocessed according to the (0,1) standardized formula, and the dimensional vibration signal is transformed into the dimensionless signal expression through Equation (18) so as to improve the sample training speed;
(18)$$x^{*}=\frac{x-x_{min}}{x_{max}-x_{min}}$$

3.The training samples are randomly set to 0, or Gaussian noise is added to realize the “damage noise” addition to simulate the fault data collected in the actual test and determine the network structure, such as the number of the SDAE input layer nodes, the number of hidden layer nodes and the number of nodes in each layer;4.The single hidden layer feedforward neural network is used as the basic model to construct multiple autoencoders, and the pseudo-inverse learning algorithm is used to train each autoencoder separately to obtain the connection weight and offset of the i-layer autoencoder. The hidden layer output of the former autoencoder is used as the input of the latter autoencoder, and the above steps are repeated to train the new autoencoder step-by-step;5.Fine-tune the parameters of the SDAE network according to the known types of faults, complete the sample feature extraction, and use the SDAE output data as the input of the support vector machine for training, diagnosis and classification.

However, When the amount of data are too large, it is easy to overfit, and other parameters need to be modified, like hyperparameter. Because the amount of data collected in this paper is not large, so it is not necessary to carry out the step in this experiment.

### 3.2. SVM Classification

After fault feature extraction, a support vector machine is used to classify the fault types, as shown in Figure 4:

The SVM learning model takes the fault features extracted by the SDAE algorithm as input. According to the trained model, the fault type is judged. If the fault is not a single mode, the concrete discriminant model is further input to analyze the coupling fault. The output function of fault probability in the SVM is shown in Equation (19):(19)$$\{\begin{array}{c} p(y=1|f\left(x\right)=\frac{1}{1+\exp\left(Af\left(x\right)+B\right)}\\ p(y=-1|f\left(x\right)=1-\frac{1}{exp\left(Af\left(x\right)+B\right)} \end{array}$$
where A and B are the shape parameters of the function. The final output of the SVM is the probability value of a motor fault between [0, 1].

## 4. Results

The motor fault data collected in this experiment is limited; the verification effect is not universal. To verify the feasibility and accuracy of the fault diagnosis method described in this paper, the bearing data set of Case Western Reserve University is used for simulation verification [31]. In the experiment, the SKF2605 drive end bearing of SKF2605 was selected, the sampling frequency was 12 Hz, the motor load was 0 horsepower, the speed was approximately 1797 r/min, and the single point damage of the bearing was carried out by electric spark to simulate the fault. The faults include nine kinds of single-point faults and normal working conditions with diameters of 0.1778 mm, 0.3556 mm and 0.5334 mm at the inner ring, outer ring and bearing roller, respectively. In the simulation, 400 groups of data are randomly selected from the above ten bearing states as the training set, and 60 groups of data are selected as the test set. The description of bearing fault data is shown in the Table 1 below.

The optimal parameters in the program were obtained after many experiments. Set the number of the input layer and hidden layer nodes of the stacked denoising autoencoder learning model as 250 and 150, respectively, the number of hidden layers as 3, the denoising parameter used to improve the accuracy of the algorithm to judge the damaged data, set as 0.2, and the training learning rate is used to control the convergence of the algorithm, set as 0.6, other parameters of the SDAE, such as hyperparameters, are mainly used to solve the problem that the algorithm is easy to cause overfitting. However, due to the small amount of data in this paper, the over-fitting phenomenon is not easy to occur, so such parameters are not added. Using SVM classifier in MATLAB Libsvm toolbox for simulation. In the test, SDAE was first used to extract the features of motor fault data and then input the extracted features into the SVM for classification to realize the process of fault diagnosis. In the program, RBF is selected as the kernel function of the SVM. This is because when the parameters of the RBF kernel function are adjusted to a certain value, it can be regarded as a linear kernel function, and the influence of the adjustment of the RBF parameters on the experimental results will not have a large deviation. At the same time, it can deal with nonlinear problems, which is suitable for the data dimension of this paper. After many experiments, it is found that the convergence effect can be achieved when the number of iterations is about 50, so the number of iterations is set to 60. Moreover, inputs the preprocessed data into the constructed denoising autoencoder for training. The total running time is 12 min, and the accuracy of the SDAE training is shown in Figure 5. The precision and recall of the SDAE are 7 and 7.5. With the increase of the number of iterations, the training accuracy gradually improves and shows a general convergence trend. For the RBF method, the accuracy can be significantly improved when the number of iterations is small, but the final accuracy is not as good as the method proposed in this paper. For a single autoencoder method, there is a small fluctuation in the iterative process; the stability is poor.

Select a motor for experimental data acquisition and subsequent analysis. The motor and bench used in the experiment are shown in Figure 6. By changing the number of turns of the stator winding, the short circuit fault of the permanent magnet synchronous motor can be simulated, and the negative sequence current can be used as the characteristic quantity of the inter-turn short circuit. Some motor data collected are shown in Table 2 below:

The right side of the table is the fault state, 1 is normal, 2 is the occurrence of inter-turn short circuit fault. Two hundred groups of data were used as training samples, and 100 groups were used as test samples. Figure 7 is an accurate image of PMSM fault diagnosis using the SDAE + SVM method.

RBF is also a neural network algorithm for classification [23], and here it is to compare it with the method described in this paper. Different algorithms are used to set the same network structure parameters, and the motor data are analyzed, as shown in Table 3:

It can be seen from the above table, the accuracy of the fault diagnosis classification method, which combines stacked denoising autoencoder and support vector machine is 94.2%, compared with the traditional algorithm RBF, single SVM and the autoencoder algorithm are only 86.6%, 89.1%, and 90.6%, respectively, which means SDAE has higher diagnosis accuracy in the actual training of motor; from the standard deviation, RBF, single SVM and autoencoder algorithm are 2.21, 0.89, 1.49, respectively; however, the standard deviation of the method proposed in this paper is only 0.88. This shows that the fluctuation of the diagnosis result using SDAE + SVM is more stable, and the robustness is better. Although the standard deviation of the single SVM algorithm is small, its accuracy is also lower than the method of this paper.

It can be seen from the above experiments that the RBF has a good convergence rate for PMSM fault extraction and classification of electric vehicles, but the accuracy is not very high, and the data integrity is required to be high. However, the RBF algorithm has the advantages of simple and good generalization ability, which is suitable for classification with complete data and low requirements; when using a single SVM to classify faults, although the fluctuation of accuracy is small, it still does not reach the ideal accuracy, and when the amount of data are large, a single SVM classifier does not have obvious advantages, which means it may cause a long operation time. Compared with a single SVM classifier, the accuracy of DAE for motor fault diagnosis fluctuates less and is more stable. The proposed SDAE + SVM method combines the single SVM and DAE algorithm as a new PMSM diagnosis method and increases the number of hidden layers on the basis of DAE to form the SDAE algorithm. It improves the learning effect of the program. Experiments show that the method has better accuracy and convergence speed, but the generalization performance needs to be further verified.

## 5. Conclusions

Based on the principle of stacked denoising autoencoder, this paper proposes a fault diagnosis method of electric vehicle PMSM based on the SDAE. The feature extraction of motor fault based on stacked denoising autoencoder algorithm has better generalization ability and diagnostic accuracy and has better recognition advantage in unsupervised learning. Compared with the traditional diagnosis method, this method can effectively avoid artificial error and diagnosis time; under the same network structure, compared with a single diagnosis algorithm, this method has better robustness and more stable diagnosis results. The research of this paper provides a new idea for the diagnosis of permanent vehicle magnet synchronous motor on the basis of intellectualization and provides a certain basis for the fault diagnosis of PMSM under the complex operation of vehicles in the future.

The diagnosis results show that under the same sample characteristics, compared with other classification diagnosis algorithms, the test accuracy of the SDAE for motor fault diagnosis can reach 94.2%; however, the accuracy rates of the RBF, single SVM and DAE are only 86.6%, 89.1% and 90.6%, respectively, and the standard deviation of the SDAE diagnostic algorithm is also small, only 0.88%, which means that the test results of the SDAE algorithm are relatively stable and have no big fluctuation.

However, there are still some defects in the above research: the fault experiments of permanent magnet synchronous motor are carried out in the normal physical field environment, without considering the influence of temperature, load and other factors on the motor work; in addition, because the motor fault in this paper is artificially set, the data results are limited, and it does not reach the ideal sample size, so there are still of some test limitations in statistics.

Based on the above shortcomings, this paper will study the operation and fault conditions of permanent magnet synchronous motor under a complex environment in the future, learning and analyzing the characteristics of coupling fault phenomena and carry out more experiments to expand the data sample size of the results; in the aspect of the classification algorithm, learn and use the improved SVM to analyze the diagnosis results more accurately, and continue to perfect the experimental program to improve the generalization performance and recognition advantages of th method that discussed in this paper.

## Figures and Tables

**Figure 1 entropy-23-00339-f001:**
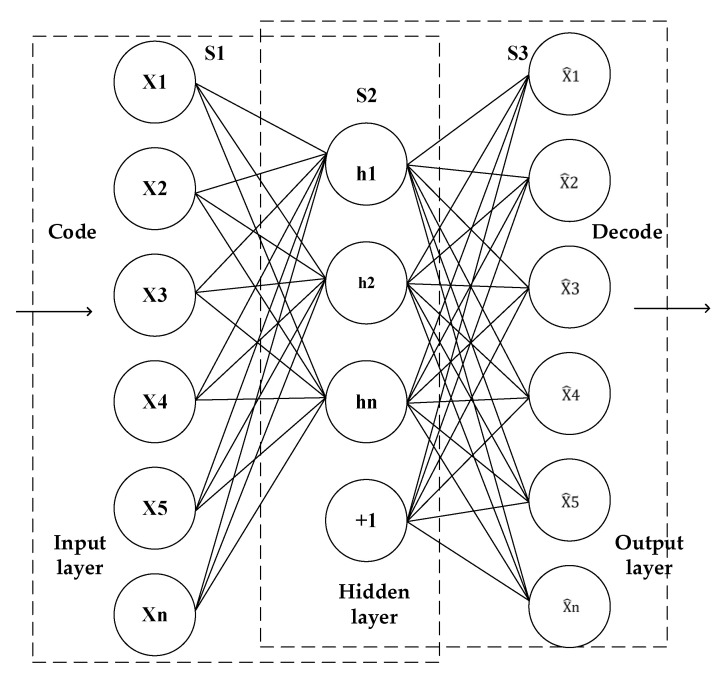
Autoencoder network structure.

**Figure 2 entropy-23-00339-f002:**
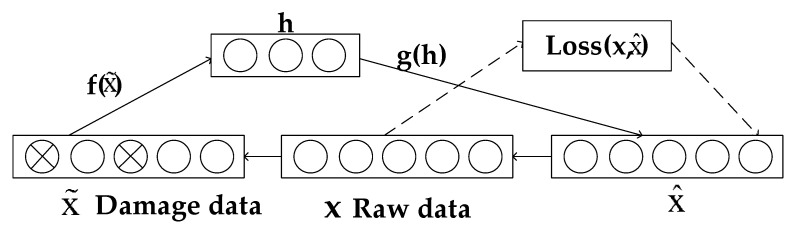
SDAE network structure.

**Figure 3 entropy-23-00339-f003:**
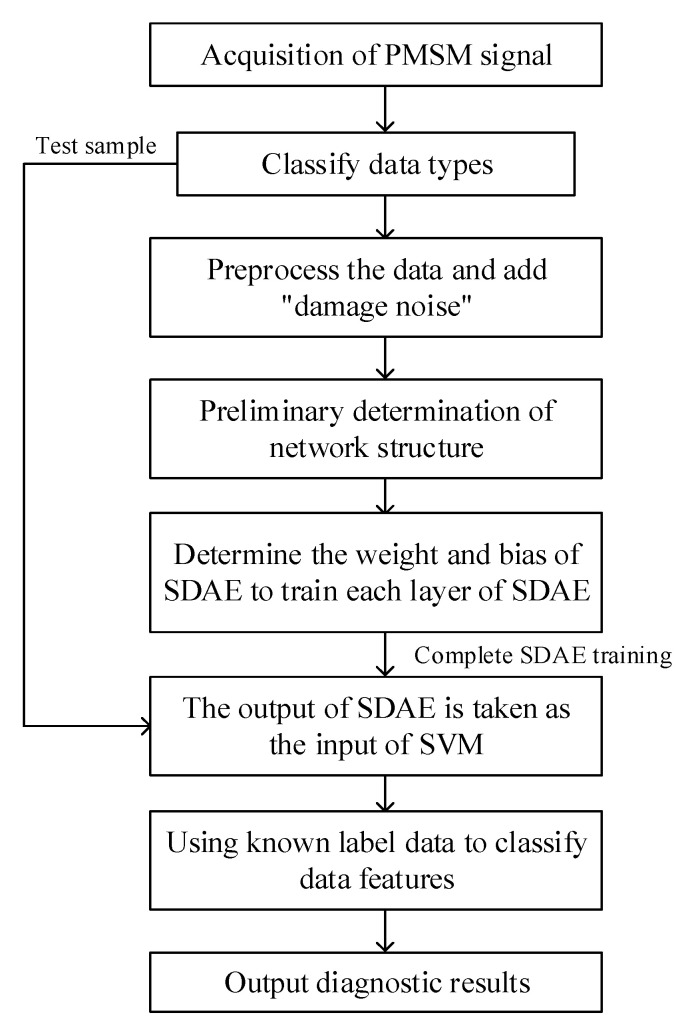
Fault diagnosis flow chart.

**Figure 4 entropy-23-00339-f004:**
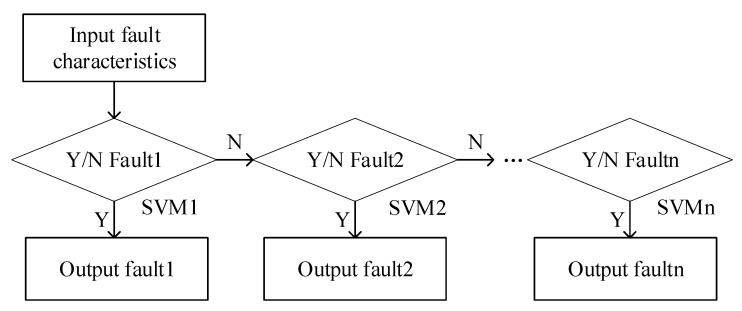
Support vector machine (SVM) classification flow chart.

**Figure 5 entropy-23-00339-f005:**
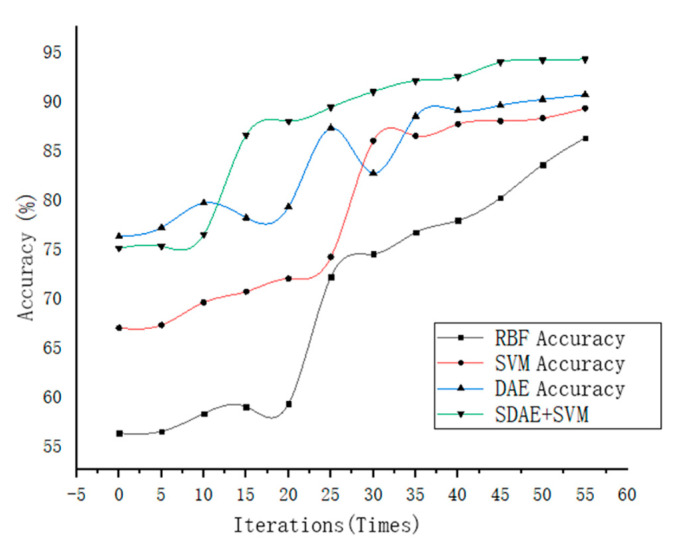
Diagnostic accuracy based on the stacked denoising autoencoder (SDAE).

**Figure 6 entropy-23-00339-f006:**
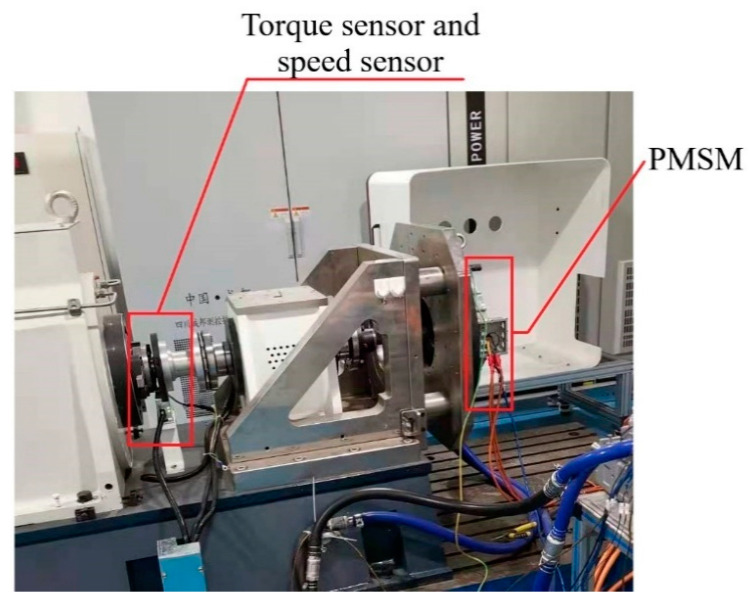
Motor bench used in the experiment.

**Figure 7 entropy-23-00339-f007:**
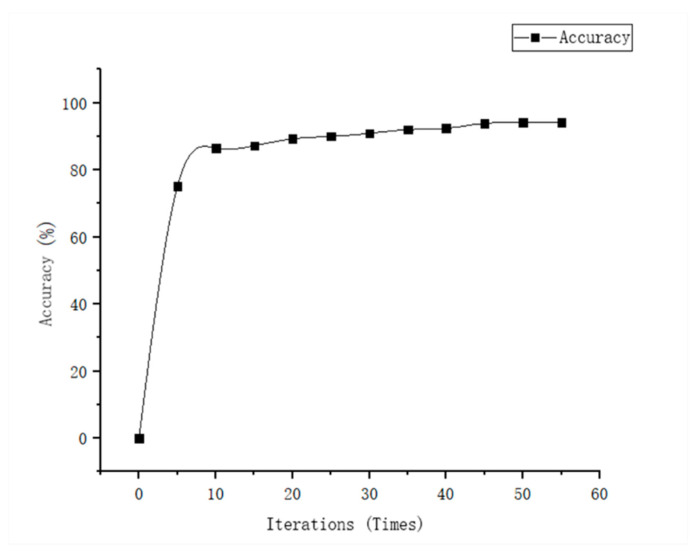
Diagnostic accuracy based on the SDAE.

**Table 1 entropy-23-00339-t001:** Bearing failure data.

Failure Mode	Fault Size/mm	Number of Samples
normal	0	400
Inner circle	0.1778	400
Inner circle	0.3556	400
Inner circle	0.5334	400
Outer ring	0.1778	400
Outer ring	0.3556	400
Outer ring	0.5334	400
Rolling element	0.1778	400
Rolling element	0.3556	400
Rolling element	0.5334	400

**Table 2 entropy-23-00339-t002:** Partial fault characteristic data.

A Phase Current	B Phase Current	C Phase Current	Negative Sequence Current	Electromagnetic Torque
1.0026	1.0086	0.9946	0.036	3.67
1.1210	1.0023	0.9934	0.064	3.77
1.1230	1.0020	0.9867	0.070	3.79
1.1339	1.0018	0.9812	0.079	3.81
1.1472	1.0007	0.9745	0.082	3.88
1.2486	0.9898	0.9658	0.182	3.92
1.5684	0.9750	0.9562	0.486	4.12
1.6982	0.9698	0.9236	0.669	4.18
1.8675	0.9672	0.8645	0.948	4.26
2.0784	0.9542	0.7996	1.072	4.40

**Table 3 entropy-23-00339-t003:** Fault test accuracy rates under different algorithms.

Classification Algorithm	Test Accuracy/%	Standard Deviation of Accuracy
RBF	86.6	2.21
SVM	89.1	0.89
DAE	90.6	1.49
SDAE + SVM	94.2	0.88

## Data Availability

The data presented in this study are available on request from the corresponding author.

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
