# Peer review of "Fault Diagnosis of Permanent Magnet Synchronous Motor Based on Stacked Denoising Autoencoder"

_entropy, 2021, doi:10.3390/e23030339_

Round 1

Reviewer 1 Report

Paper deals with important and interesting task. Authors have proposed an approach for fault feature extraction for motors based on the principle of Stacked Denoising Autoencoder (SDAE) combined with SVM classifier.

Paper has scientific novelty and GREAT! practical value.

It has a logical structure, all necessary sections. Paper is technically sound. Experimental section is very good.

Proposed approach is logical, results are clear.

Suggestions:

  1. Introduction section should be extended using point-by-point the main contributions of this paper
  2. It would be good to add the reminder of this paper
  3. Authors should add a Related works section. It should have an analysis of the existing ensembles. Authors should use these papers: DOI: 10.1109/DASA51403.2020.9317124 , DOI: 10.1016/j.jestch.2020.10.005 , DOI:10.3390/s20092625 among others
  4. Please argue your choice of the classifier for your approach. Why SVM?
  5. Authors should add all optimal parameters for all investigated methods
  6. Please provide the procedures for choosing SVM's optimal parameters in the term of the proposed method. What kernel should be used, how much iterations and others...
  7. Please provide the training time for all investigated methods
  8. It would be good to see Precision and Recall measures because total accuracy isn't a perfect indicator.
  9. Conclusion section should be extended using: 1) numerical results obtained in the paper; 2) limitations of the proposed approach; 3) prospects for the future research.
  10. Please add more references. Some of the existing references are outdated. Please fix it using 3-5 years old papers in high-impact journals.

Other suggestions

  1. Fig. 5 and 6 has a bad quality. Please fix it.

Author Response

Thank you for your rigorous comments on Fault Diagnosis of Permanent Magnet Synchronous Motor Based on Stacked Denoising Autoencoder. Your constructive suggestions and criticisms are of great significance to the revision of this paper and our future research. We have made the following amendments to the article according to the valuable comments of you:

  1. Introduction section should be extended using point-by-point the main contributions of this paper.

Response: In the introduction, we supplement the logical writing order of this paper in page 3, line 95-102.

  1. It would be good to add the reminder of this paper.

Response: We have added some relevant paragraphs to the conclusion of the article, in page 12, line 338-341. But we are not sure whether it is consistent with the modification of the opinion

  1. Authors should add a Related works section. It should have an analysis of the existing ensembles. Authors should use these papers: DOI: 10.1109/DASA51403.2020.9317124, DOI: 10.1016/j.jestch.2020.10.005, DOI:10.3390/s20092625, among others.

Response: In the paper, we have added those related works as required in page 2, line 60-62.

  1. Please argue your choice of the classifier for your approach. Why SVM?

Response: Compared with the deep learning classification method, SVM is easy to operate in the program, and it can get higher accuracy without using a lot of data, in this paper, the experimental motor data is just suitable for SVM classification, so this classification method is selected. We also further elaborate the reason in page 6, line 186-191.

  1. Authors should add all optimal parameters for all investigated methods.

Response: Other parameters of SDAE, such as hyperparameters, are mainly used to solve the problem that the algorithm is easy to cause over fitting. However, due to the small amount of data in this paper, the over fitting phenomenon is not easy to occur, so such parameters are not added. And we also explain this situation in page 9, line 282-285.

  1. Please provide the procedures for choosing SVM's optimal parameters in the term of the proposed method. What kernel should be used, how much iterations and others.

Response: According to the editor's suggestion, we discuss the choice of kernel function and iteration number in page 9, line 286-293.

  1. Please provide the training time for all investigated methods.

Response: We did not set a fixed running time for the program. After the re experiment, the total training time is 12 mins. And we added this data in page 9, line 294.

  1. It would be good to see Precision and Recall measures because total accuracy isn't a perfect indicator.

Response: In the process of experiment, we found that the recall and precision of the proposed method did not reach the ideal value, so we did not take this as the basis for evaluation, but we still added the value as suggestion in page 9, line 295.

  1. Conclusion section should be extended using: 1) numerical results obtained in the paper; 2) limitations of the proposed approach; 3) prospects for the future research.

Response: In the conclusion part, the author has carried on the supplementary explanation according to numerical results obtained in the paper, limitations of the proposed approach, and prospects for the future research.In page 12, line 342-361.

  1. Please add more references. Some of the existing references are outdated. Please fix it using 3-5 years old papers in high-impact journals.

Response: We have replaced the literature 1, 11, 17 and 24 with several more recent articles.

  1. Fig. 5 and 6 has a bad quality. Please fix it.

Response: We tried to improve the quality of the photos in page 10 and 11.

Reviewer 2 Report

Authors try to solve a very important task for inductry. Authors proposed AI-based approach for fault feature extraction for motor.

Such work has big practical value.

The proposed approach is clear.

What should be improved:

  1. There are no scientific novelty in the end of the first section. Please add it to this paper.
  2. It is unclear why authors used SVM as a basic classificator in their model. There are a lot of improved version of the SVM (10.5815/ijisa.2018.09.05) that are mode precisioly than the basic SVM
  3. I cant find the training time F-measure for the proposed approach. It should be added to this paper
  4. Authors should add Discusion section
  5. Conclusion section is very short. Please fix it.

Author Response

Thank you for your rigorous comments on Fault Diagnosis of Permanent Magnet Synchronous Motor Based on Stacked Denoising Autoencoder. Your constructive suggestions and criticisms are of great significance to the revision of this paper and our future research. We have made the following amendments to the article according to the valuable comments of you:

  1. There are no scientific novelty in the end of the first section. Please add it to this paper.

Response: We added some relevant paragraphs at the end of the first part, in page 2 and 3, line 88-94.

  1. It is unclear why authors used SVM as a basic classificator in their model. There are a lot of improved version of the SVM (10.5815/ijisa.2018.09.05) that are mode precisely than the basic SVM.

Response: (1) SVM is easy to operate in the program, and it can get higher accuracy without using a lot of data, which is suitable for the situation of less motor data collected in this paper, and we have also made a supplementary explanation to this situation in page 6, line 186-191.

(2) Due to the small amount of motor fault data obtained in this experiment, the ordinary SVM classifier is enough to adapt to achieve the ideal effect, so the improved SVM classification method is not considered in the experiment for the time being. In the next research, we will further learn the improved SVM method, and follow up the modification of the data acquisition method of the fault motor, deeply study the complex problems such as coupling fault, and continue to improve the methods proposed in this paper.

  1. I cant find the training time F-measure for the proposed approach. It should be added to this paper.

Response: We did not set a fixed running time for the program. After the re experiment, the total training time is 12 mins. And we added this data in page 9, line 294.

  1. Authors should add Discussion section.

Response: According to the suggestion, we have supplemented some numerical analysis to the experimental results in page 11, line 318-329.

  1. Conclusion section is very short. Please fix it.

Response: At the end of the paper, we have made some supplements as suggested from the following three aspects: numerical analysis of experimental results data, the limitations of the paper and the future research directions in page 12, line 324-361.

Reviewer 3 Report

The manuscript presents an interesting proposal for the diagnosis of PMSMs and the proposed methodology may be valid and a suitable solution. However, the manuscript has many shortcomings and would need extensive rewriting before possible publication.

  • Several times vehicles are mentioned, but is that relevant, does it have anything to do with the rest of the article?
  • It is not clear why the procedures are used. It should be explained in more detail why the autoencoder is used before applying a classifier, what advantages are obtained with respect to applying only one of the two procedures.
  • Line 187: qualify the statement that it is the coupling effect of the electromagnetic field that causes motor faults to appear.
  • Line 190: qualify the statement “slow speed of motor fault feature extraction”
  • Line 199:” In the Stacked Denoising Autoencoder part, the data processing generally includes the following steps” Why “generally”? When does not include those steps?
  • Line 227: what do you mean with “The first mock exam”?
  • Further explanation of the parameters used would be required, e.g.: “Set the number of input layer and hidden layer nodes of the Stacked Denoising 248 Autoencoder learning model as 250 and 150 respectively, the number of hidden layers as 249 3, the denoising parameter as 0.2, and the training learning rate as 0.6.”
  • In the introduction, windings are highlighted but then bearings are analyzed, why?
  • Line 250: “SVM uses Libsvm toolbox in MATLAB for experimental simulation, and inputs the preprocessed data into the constructed Denoising Autoencoder for training” What does SVM have to be with “experimental simulation”? Is SVM applied prior to DA?
  • Line 254, Why does RBF appear here, is it a comparison with other methods? That hasn't been explained
  • Línea 260: “Select a motor for experimental data acquisition and subsequent analysis “this is unrelated to the above
  • Line 261: why are you suddenly talking about stator windings here when you were dealing with bearings?
  • Separate the figures from the units. For example, "0.1778mm".

Author Response

Thank you for your rigorous comments on Fault Diagnosis of Permanent Magnet Synchronous Motor Based on Stacked Denoising Autoencoder. Your constructive suggestions and criticisms are of great significance to the revision of this paper and our future research. We have made the following amendments to the article according to the valuable comments of you:

  1. Several times vehicles are mentioned, but is that relevant, does it have anything to do with the rest of the article?

Response: This is because the research of this paper is based on vehicle permanent magnet synchronous motor, and we also explained it in the introduction in page 2, line 85-87.

  1. It is not clear why the procedures are used. It should be explained in more detail why the autoencoder is used before applying a classifier, what advantages are obtained with respect to applying only one of the two procedures?

Response: The main purpose of SDAE algorithm is to extract the features of the collected PMSM data, which is equivalent to label the data and facilitate the subsequent SVM to classify it. And the SDAE used in this paper adds noise processing in the data input, which has better adaptability for the actual incomplete data. We also supplement this part in page 2, line 88-91. The experiment in this paper adopts the way of combining two methods and using a single method will make the experiment unable to complete.

  1. Line 187: qualify the statement that it is the coupling effect of the electromagnetic field that causes motor faults to appear?

Response: It is not because the effect of electromagnetic field will lead to the motor failure in the process of operation, but after the motor has a tendency to damage, the effect of electromagnetic field will make the fault happen faster and more obvious. We also made changes in page 6, line 211-213.

  1. Qualify the statement “slow speed of motor fault feature extraction”.

Response: The original feature extraction method uses artificial discrimination to operate, and the time of artificial discrimination is obviously longer than that of intelligent algorithm.

  1. Line 199:” In the Stacked Denoising Autoencoder part, the data processing generally includes the following steps” Why “generally”? When does not include those steps?

Response: Because the amount of data collected in this paper is small, there is no need to add hyperparameters in SDAE classification. When the amount of data is too large, it is easy to over fit, and other parameters need to be modified. In view of this situation, we have also made a corresponding supplement in page 8, line 247-249.

  1. Line 227: what do you mean with “The first mock exam”?

Response: The purpose of this is to show that in the process of experiment, the output result of fault features extracted by SDAE is used as the input of SVM classifier. We have revised the sentence in page 8, line 256.

  1. Further explanation of the parameters used would be required, e.g.: “Set the number of input layer and hidden layer nodes of the Stacked Denoising 248 Autoencoder learning model as 250 and 150 respectively, the number of hidden layers as 249 3, the denoising parameter as 0.2, and the training learning rate as 0.6.”

Response: The input layer and hidden layer are introduced in page 5, line 159-176; and the meaning of denoising parameter and training learning are explained as suggested in page 9, line 280-282.

  1. In the introduction, windings are highlighted but then bearings are analyzed, why?

Response: Because the motor fault data collected in this experiment is limited, the verification effect is not universal; in order to verify the feasibility and limitation of the method proposed in this paper, the bearing data is used to verify firstly. We make a supplement in page 8, line 265.

  1. Line 250: “SVM uses Libsvm toolbox in MATLAB for experimental simulation, and inputs the preprocessed data into the constructed Denoising Autoencoder for training” What does SVM have to be with “experimental simulation”? Is SVM applied prior to DA?

Response: This sentence is to illustrate that we use the SVM classifier in MATLAB toolbox, and using SVM in MATLAB to classify data is a part of the simulation experiment. In the experimental method of this paper, we need to firstly use SDAE to extract the features of motor fault data, and then input the extracted features into SVM for classification to realize the process of fault diagnosis. So, it is SDAE that applied prior to SVM.

  1. Line 254, Why does RBF appear here, is it a comparison with other methods? That hasn't been explained

Response: RBF is also a neural network algorithm for classification. Here we propose RBF is to compare the proposed method with other methods. We make a supplement in page 11, line 314.

  1. Line 260: “Select a motor for experimental data acquisition and subsequent analysis” this is unrelated to the above

Response: Because the above part about the bearing is to verify the feasibility and effectiveness of the method proposed in this paper, and this paper is mainly a new experimental method for motor fault, because the amount of motor fault data collected in the experiment is less, so we use the bearing data to verify the feasibility of the method. After the bearing data analysis, the feasibility and effectiveness of this method can be verified, so the motor is selected for experimental analysis.

  1. Line 261: why are you suddenly talking about stator windings here when you were dealing with bearings?

Response: This paper is mainly a new experimental method for motor fault, because the amount of motor fault data collected in the experiment is less, so we use the bearing data to verify the feasibility of the method. After the bearing data analysis, the feasibility and effectiveness of this method can be verified, so the motor is selected for experimental analysis.

  1. Separate the figures from the units. For example, "0.1778mm".

Response: We have revised the corresponding parts in page 8, line 269, 270, 272 and 273.

Round 2

Reviewer 1 Report

Paper can be acceptesd

Author Response

Dear reviewer, thank you again for your valuable suggestions on Fault Diagnosis of Permanent Magnet Synchronous Motor Based on Stacked Denoising Autoencoder. Your precious comments on this article will be of great significance to the publication and future research of our paper.

Reviewer 2 Report

Accepted

Author Response

(The authors gave the same response as above.)

Reviewer 3 Report

Please, see attached file

Author Response

Dear reviewer, thank you again for your valuable suggestions on Fault Diagnosis of Permanent Magnet Synchronous Motor Based on Stacked Denoising Autoencoder. Your precious comments on this article will be of great significance to the publication and future research of our paper. In response to your comments, we have made the following replies:

1. Several times vehicles are mentioned, but is that relevant, does it have anything to do with the rest of the article?

Response: This is because the research of this paper is based on vehicle permanent magnet synchronous motor, and we also explained it in the introduction in page 2, line 85-87.

OK, but, does the case study correspond to a vehicle motor?

Respons: Yes, the study is correspond mainly to the Electric Vehicle motor. Compared with the general PMSM, the vehicle PMSM has the characteristics of small volume, high efficiency and high power density. However, most of the motors work in the closed and narrow complex harsh environment, with complex working conditions and large load changes. Therefore, the vehicle motor is more prone to a variety of faults. And we make a supplementary description in page 1 and 2, line 30-42.

2. It is not clear why the procedures are used. It should be explained in more detail why the autoencoder is used before applying a classifier, what advantages are obtained with respect to applying only one of the two procedures?

Response: The main purpose of SDAE algorithm is to extract the features of the collected PMSM data, which is equivalent to label the data and facilitate the subsequent SVM to classify it. And the SDAE used in this paper adds noise processing in the data input, which has better adaptability for the actual incomplete data. We also supplement this part in page 2, line 88-91. The experiment in this paper adopts the way of combining two methods and using a single method will make the experiment unable to complete.

OK

3. Line 187: qualify the statement that it is the coupling effect of the electromagnetic field that causes motor faults to appear?

Response: It is not because the effect of electromagnetic field will lead to the motor failure in the process of operation, but after the motor has a tendency to damage, the effect of electromagnetic field will make the fault happen faster and more obvious. We also made changes in page 6, line 211-213.

I still think that it would not be clear for the reader what the authors mean with “the coupling effect of electromagnetic field will make 212 the faults happen faster and more obvious”.

Respons: What we want to express here is when the motor has a slight short circuit fault, if it is not checked and repaired in time, it will lead to the increase of the motor operating temperature. That will cause the change of the working temperature field, which will increase the degree of the motor inter-turn short circuit fault, and even cause the demagnetization of the permanent magnet. The change of the working magnetic field caused by demagnetization will aggravate the degree of short circuit and other faults. And we make a supplementary description in page 6 and 7, line 214-220.

4. Qualify the statement “slow speed of motor fault feature extraction”.

Response: The original feature extraction method uses artificial discrimination to operate, and the time of artificial discrimination is obviously longer than that of intelligent algorithm.

But it is still not clear in the manuscript what the authors mean with that sentence.

Respons: In the traditional manual diagnosis of fault data extraction features, the accuracy of the diagnosis results depends on the technical level and practical experience of the diagnosis personnel, its self-learning ability is weak and the intelligent level is low; while the fault diagnosis method of signal analysis has strong dependence on data and poor generalization ability. Compared with this method, the proposed method has faster speed in feature extraction of motor fault data. And we have made supplement as suggested in page 7, line 221-225.

5. Line 199:” In the Stacked Denoising Autoencoder part, the data processing generally includes the following steps” Why “generally”? When does not include those steps?

Response: Because the amount of data collected in this paper is small, there is no need to add hyperparameters in SDAE classification. When the amount of data is too large, it is easy to over fit, and other parameters need to be modified. In view of this situation, we have also made a corresponding supplement in page 8, line 247-249.

OK

6. Line 227: what do you mean with “The first mock exam”?

Response: The purpose of this is to show that in the process of experiment, the output result of fault features extracted by SDAE is used as the input of SVM classifier. We have revised the sentence in page 8, line 256.

OK

7. Further explanation of the parameters used would be required, e.g.: “Set the number of input layer and hidden layer nodes of the Stacked Denoising 248 Autoencoder learning model as 250 and 150 respectively, the number of hidden layers as 249 3, the denoising parameter as 0.2, and the training learning rate as 0.6.”

Response: The input layer and hidden layer are introduced in page 5, line 159-176; and the meaning of denoising parameter and training learning are explained as suggested in page 9, line 280-282.

But it is still not clear why those specific values have been chosen.

Respons: In the program of intelligent algorithm, these parameters are not determined as the final best results at one time. Instead, experiments are carried out in the range of parameters given by existing researchers to find the optimal parameters. And relatively good values are found after many operations. We also supplement the reason in page 9, line 289.

8. In the introduction, windings are highlighted but then bearings are analyzed, why?

Response: Because the motor fault data collected in this experiment is limited, the verification effect is not universal; in order to verify the feasibility and limitation of the method proposed in this paper, the bearing data is used to verify firstly. We make a supplement in page 8, line 265.

OK

9. Line 250: “SVM uses Libsvm toolbox in MATLAB for experimental simulation, and inputs the preprocessed data into the constructed Denoising Autoencoder for training” What does SVM have to be with “experimental simulation”? Is SVM applied prior to DA?

Response: This sentence is to illustrate that we use the SVM classifier in MATLAB toolbox, and using SVM in MATLAB to classify data is a part of the simulation experiment. In the experimental method of this paper, we need to firstly use SDAE to extract the features of motor fault data, and then input the extracted features into SVM for classification to realize the process of fault diagnosis. So, it is SDAE that applied prior to SVM.

OK .But it is still not clear in the manuscript. The authors should further explain this issue in the text.

Respons: We have further explained this issue “Using SVM classifier in MATLAB Libsvm toolbox in MATLAB for experiment. In the test, SDAE was firstly used to extract the features of motor fault data, and then input the extracted features into SVM for classification to realize the process of fault diag-nosis.” in page 9, line 297-300 as suggested. Besides, our expression here is not very appropriate. The SVM classifier in Matlab toolbox is used for program verification rather than simulation experiment, and we also modified the expression.

10. Line 254, Why does RBF appear here, is it a comparison with other methods? That hasn't been explained

Response: RBF is also a neural network algorithm for classification. Here we propose RBF is to compare the proposed method with other methods. We make a supplement in page 11, line 314.

But it should be made clear in the text that it is for comparison reasons.

Respons: We have supplemented that RBF here is to compare with the method described in the paper in page 11, line 329-330.

11. Line 260: “Select a motor for experimental data acquisition and subsequent analysis” this is unrelated to the above

Response: Because the above part about the bearing is to verify the feasibility and effectiveness of the method proposed in this paper, and this paper is mainly a new experimental method for motor fault, because the amount of motor fault data collected in the experiment is less, so we use the bearing data to verify the feasibility of the method. After the bearing data analysis, the feasibility and effectiveness of this method can be verified, so the motor is selected for experimental analysis.

But it should be made clear in the text.

Respons: We have made supplement that to verify the feasibility and accuracy of the fault diagnosis method described in this paper, the bearing data set is used for simulation verification as suggested in page 9, line 276-279.

12. Line 261: why are you suddenly talking about stator windings here when you were dealing with bearings?

Response: This paper is mainly a new experimental method for motor fault, because the amount of motor fault data collected in the experiment is less, so we use the bearing data to verify the feasibility of the method. After the bearing data analysis, the feasibility and effectiveness of this method can be verified, so the motor is selected for experimental analysis.

But it should be made clear in the text.

Respons: We have made supplement as suggested that the motor fault data collected in this experiment is limited and the bearing data is used to verify the feasibility of the method in page 9, line 276-279.

13. Separate the figures from the units. For example, "0.1778mm".

Response: We have revised the corresponding parts in page 8, line 269, 270, 272 and 273.

OK
